# AMR-Assisted Order Picking: Models for Picker-to-Parts Systems in a Two-Blocks Warehouse

**Giulia Pugliese [1], Xiaochen Chou [1], Dominic Loske [2], Matthias Klumpp [2,3] and Roberto Montemanni [1,*]**

[1] Department of Sciences and Methods for Engineering, University of Modena and Reggio Emilia, 42122 Reggio Emilia, Italy
[2] Retail Logistics Lab, Department of Business Administration, Georg-August University of Göttingen, 37073 Göttingen, Germany
[3] Fraunhofer Institute for Material Flow and Logistics (IML), 44227 Dortmund, Germany
[*] Correspondence: roberto.montemanni@unimore.it

**Abstract:** Manual order picking, the process of retrieving stock keeping units from their storage location to fulfil customer orders, is one of the most labour-intensive and costly activity in modern supply chains. To improve the outcome of order picking systems, automated and robotized components are increasingly introduced creating hybrid order picking systems where humans and machines jointly work together. This study focuses on the application of a hybrid picker-to-parts order picking system, in which human operators collaborate with Automated Mobile Robots (AMRs). In this paper a warehouse with a two-blocks layout is investigated. The main contributions are new mathematical models for the optimization of picking operations and synchronizations. Two alternative implementations for an AMR system are considered. In the first one handover locations, where pickers load AMRs are shared between pairs of opposite sub-aisles, while in the second they are not. It is shown that solving the mathematical models proposed by the meaning of black-box solvers provides a viable algorithmic optimization approach that can be used in practice to derive efficient operational plannings. The experimental study presented, based on a real warehouse and real orders, finally allows to evaluate and strategically compare the two alternative implementations considered for the AMR system.

**Keywords:** AMR-assisted; hybrid order picking system; picker-to-parts system; mixed integer linear programming; optimization; two-blocks warehouse; case study

## 1. Introduction

The optimization of order picking process in the warehouses is a well-studied problem in literature; retrieving stock keeping units to fulfil orders, in fact, is among the most critical and expensive activities [1]. In modern warehouses, the automated component is increasingly replacing manual material handling tasks, with the aim of minimizing the time spent by the human operator for non-value-added activities. For this reason, the order picker is often accompanied by an automated cart (typically AMR or AGV, Automated Guided Vehicles) in its activity of picking stock keeping units [2].

In this paper, we study a particular hybrid order picking system, defined AMR-assisted picker-to-parts system [2], according to which every human operator, assigned to a specific aisle of the warehouse, manually picks the articles of its zone, and then load them onto an AMR in a dedicated location at the end of each corridor [2]. The AMR, after having visited all the corridors and collected all the items, takes care of transporting those from the storage area to the depot.

The contribution of our study is to specialize known mathematical programming models to a given warehouse layout based on two-blocks of sub-aisles, with the aim of planning picking operations and the coordination between pickers and AMRs in an optimal way. A second contribution is represented by the comparison between alternative

implementation of the AMR system in the given context, with an evaluation of the efficiency in terms of minimization of the total tardiness. Our mathematical models are finally applied to a real-world dataset and in close cooperation with a German brick-and-mortar grocery retailer.

The paper is structured as follow: in Section 2 we review the literature on AMR-assisted order picking; in Section 3 we describe the characteristics of the warehouse subject of our investigation; in Section 4 we define two possible implementation of the AMR-assisted Picker-to-Parts system for the given warehouse configuration and we present mathematical models; numerical experiments on the impact of speed and size of the AMR fleet into efficiency are carried out in Section 5, together with a comparison of the two possible implementations of the AMR system; conclusions are finally drawn in Section 6.

## 2. Literature Review

The activity of order picking, that is the manual or automated withdrawal of stock keeping units from storage locations to satisfy customer demand, is a frequently object of study in the literature [3,4]. It is in fact among the most labor-intensive and costly activities in any material handling system [1]. In traditional picker-to-parts manual warehouses, the items, are collected by human operators (order pickers) who operate by moving from one shelf to another [2].

A common configuration for the management of order picking activities in the warehouse consists in its subdivision in zones (we speak in this case of Zone Picking [5]). This strategy is particularly efficient to reduce the risk of congestion and improve picking performance, both in automated and manual warehouses [6]. On the other hand, once the collection operation has been completed, there is the need to group together items of the same order, still spread over several areas; this last aspect can be a significant disadvantage. There are two different approaches in Zone Picking systems: parallel (synchronized) and sequential (progressive) [6]. In the first case, pickers employed in different areas of the warehouse simultaneously collect items of the same order and the items are put together once each picker has completed her/his picking process. In the second case, however, after a picker has collected all the items of a certain order in her/his area, those are transferred to the next area, typically by means of conveyors; in the situation described, therefore, items of the same order are already grouped together. Several authors have studied the optimization of a warehouse organized in zones [7]. The choice of the optimal number of zones to divide a manual warehouse was analyzed in [8,9], with the aim of minimizing the collection time. The authors of [10], in addition, investigated the problem of assigning robots to different zones, demonstrating that the processing time of an order can be reduced by a third, sharing robots among them. Finally, the authors of [11] analyzed the relationship between the storage activities, Order Batching, Zone Picking and Routing, demonstrating that, for greater efficiency, all the steps indicated should be managed in an integrated manner. The studies mentioned above refer to systems in which the zoning is static and therefore not modifiable during the picking operations.

The research of most of authors focused on the minimization of the crossed total distance or of the time spent from the operator in its activity of retrieving orders from the warehouse and other environments (see, for example, [12–14]). From other authors, instead, a solution approach has been proposed, aimed at the minimization of the total delay (tardiness). In [15] the total delay is defined as the positive difference between the completion time of an order and its due date. Orders, in fact, must be completed within set time limits, in order not to cause delays in production or delivery to the final customer. The authors listed below have studied the problem of the minimization of the total delay in various contexts of warehouse. The authors of [16] considered the JOBASP (Joint Order Batching, Assignment and Sequencing Problem) for the case of a single picker in an automated warehouse. The same problem was treated in [17], but in the case of a non-automated warehouse. Subsequently, in [18] the analysis is extended to the case of multiple pickers, and heuristic approaches (see [19]) are employed for large instances.

Scholz et al. [20], then, proposed an integrated solution between the problems of Order Batching, Batch Sequencing and Picker Routing, aimed at minimizing the total delay. Finally, Žulj et al. [21], presented a solution for a zoned warehouse where human operators are assisted by AMR automated carts.

In recent years, automated components have become preponderant in warehouses, with the aim of speeding up operations and employing the working time of the human operators more efficiently [22]. In fact, while the investment costs for a manual warehouse are low, on the other hand the operator spends a considerable part of his working time moving from one shelving to another, and this is not efficient [23]. A possible solution to reduce this non-value-added fraction of time is to equip a human operator with an automated guided vehicle, typically an AGV (Automated Guided Vehicle) [24] or an AMR (Automated Mobile Robot) cart [25]. The main difference between these two types of automated vehicles lies in the fact that an AGV follows predefined routes, while an AMR, on the contrary, has some intelligence and can recalculate the route in case of obstacles or congestions. AMR-assisted Order Picking has many advantages, including reduced walking time for the operator, increased performance, easy implementation, paired with higher reliability and flexibility. On the other hand, employing automated vehicles requires higher investments, both in terms of startup costs and periodical maintenance [21].

Studies on optimized AMR-assisted Picker-to-Parts systems have started appearing in the literature only recently (see [26]). The authors of [21] studied an AMR-assisted Picker-to-Parts system, with the aim of minimizing the total tardiness of orders with respect to the due date. In their study, a standard single block rectangular warehouse, consisting of parallel aisles and with items stored on both sides, is analyzed. The corridors can be accessed through perpendicular main corridor placed frontally, as depicted in Figure 1. The study relies on a clear subdivision in zones of the warehouse: each aisle represents a distinct zone with a single picker assigned to it. The picker collects the items in the aisle of competence and transports them to the respective handover location, located at the crossing of the aisle with the main corridor. The handover locations are visited by AMRs, that are loaded with the picked articles and then transport the items from the picking areas to a designed depot, where items are further processed toward packing and delivering. Once the items are unloaded from the AMR, the batch is considered completed from the picking-optimization viewpoint, the AMR is equipped with new containers to collect items and can start processing a new batch.

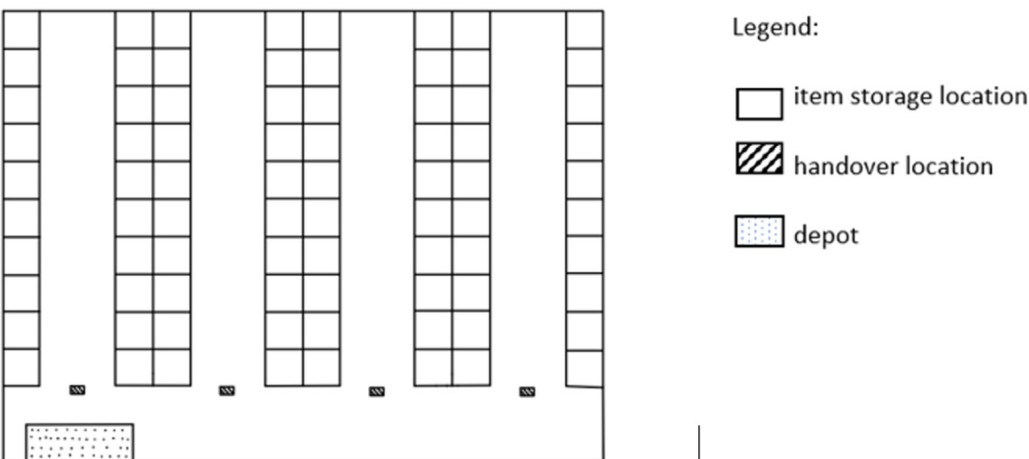

**Figure 1.** Warehouse with standard layout.

Regarding the picking activity, a synchronized Zone Picking strategy [3] is considered, according to which the items of the same batch are collected simultaneously by different pickers in different zones. If the items of a batch are stored in different aisles, as a result,

several operators are involved in retrieving of them. The quality of the final solution is measured in relation to the minimization of the total delay.

The research contributions of the study developed in [21] include the formulation of Mixed Integer Programming (MIP) for the layout depicted in Figure 1. Note that using MIPs to model and solve layout problems is a common practice (see, for example, [27]).

The models discussed in the remainder of the paper are modifications to such a model aiming at accommodating a two-blocks layout configuration and some other peculiarities emerging from the real application we consider.

### 3. The Two-Blocks Warehouse Case Study

The real-world context under study is that of a German company operating in the large-scale retail and tourism sectors. Products (food, drugs, clothing, housewares, cosmetics, consumer electronics, etc.) are stored in several warehouses. The warehouses are served by operators walking between aisles, carrying a cart in which they temporarily place the items collected. An order consists of the set of items requested by a supermarket. The Company is interested in the impact of the implementation of an AMR-assisted Picker-to-Parts system to a real-world warehouse characterized by two blocks, as depicted in Figure 2.

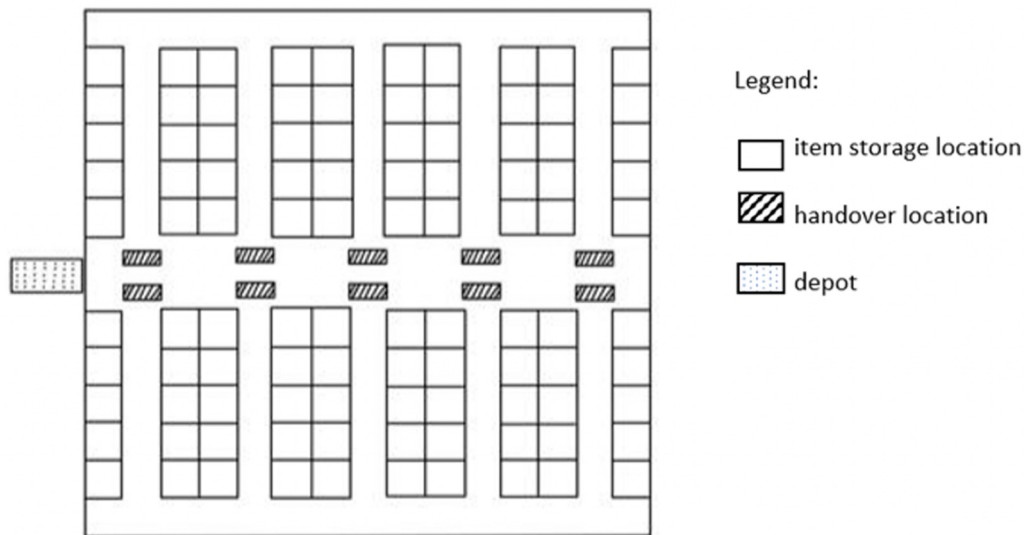

**Figure 2.** Warehouse with a two-blocks layout.

In the context of the application we consider, the problem of Order Batching is solved in an earlier stage, conversely in the case study illustrated in [21], where the assignment of customer orders to different batches is part of the optimization problem. Consequently, also the restrictions made in [21] about the relations between orders and batches are drop and the model can be simplified consequently, by reducing the number of variables and constraints. However, note that the problem of Batch Sequencing remains part of the optimization problem: knowing the number of AMR and the batches to be processed, we search for the optimal assignment and sequencing of the batches to the carts. The rest of the settings remain the same.

The AMRs begin their journey from the depot and visit the handover locations along the central corridor associated with the aisles containing batch items that the cart itself is processing, eventually turning back when the right-most aisle containing an order is reached. The depot is in front of the left-most corridor. Note that other locations are possible for depot, nut changing the location has a negligible influence on the average travel time [25]. The picker of each corridor, once returned to the respective handover location with the collected items, can load the cart, which, once the collection of all the items is finished, returns to the depot. The picker cannot start loading items until the cart has reached the handover location of its corridor. As soon as it is returned to the starting

point, the cart is downloaded from all the items in the batch, which can then be considered completed; at this point the AMR can begin the collection of the next batch.

Previous studies in the literature have shown that the presence of a central corridor in the warehouse may lead to a decrease of the travel time required by the operator [28]. In the case study considered in this paper, we have each human operator retrieving the items of a specific sub-aisle.

## 4. Implementations of an AMR-Assisted Picker-to-Parts System

In this section, two different possible implementations of an AMR-assisted Picker-to-Parts system, one with a separated handover location for each sub-aisle and one where each pair of coupled sub-aisles share a common handover location, that can be visited by the AMR either when traveling to the right or to the left, in its way back to the depot.

For each one of the two implementations a mathematical programming model inspired by the one discussed in [21] is proposed. Note that the resulting models end up being substantially different from the one discussed in [21], due to the layout (two blocks instead of one) and especially due to the application-specific settings mentioned in Section 3. Using

### 4.1. Separated Handover Locations for Sub-Aisles

In the first implementation, we consider a warehouse with two blocks of parallel aisles, with a handover location assigned to each sub-corridor. Each sub-aisle is assigned a specific order picker, who can load the items collected exclusively in the handover location mentioned above. Both the handover locations of each aisle are placed in the perpendicular main corridor, as depicted in Figure 2.

Each AMR, starting from the depot located on the left of the middle aisle, visits the handover locations of the sub-corridors clockwise. According to this configuration, the AMR first visits the corridors of the upper block and, later, those of the lower block. Once the tour is over, the AMR returns to the starting depot. Note that the AMR can eventually turn back to the depot when the right-most aisle containing an order is reached. If the number of aisles per block is n, the warehouse will consist of $2n$ sub-aisles and $2n$ handover locations, each one operating for a single sub-aisle.

### 4.1.1. A Mixed Integer Linear Programming Formulation

The warehouse object of study (Figure 2) consists of $n$ corridors and $2n$ sub-aisles, and each sub-aisle is associated with a private handover location. The set $V = \{0, 1, 2 \ldots, n, \ldots, 2n, 2n + 1\}$ denotes warehouse sub-aisles of the $n$ main aisle: depot instances are represented virtually by aisles 0 and $2n + 1$. The relevant sub-aisles for each batch are defined by the set $C_b$. It always includes the depot, which is the starting and arrival point of the AMR on each of its tours. The set of customer orders and batches are indicated respectively with $O$ and $B$. The composition of the batches is known through the binary indicator $v_{bo}$, which takes value 1 if the order $o$ is in batch $b$, 0 otherwise. The AMR fleet is modelled by set $A = \{1, 2, \ldots, m\}$. The set of tours (for each AMR) to which batches can be assigned is represented by set $K$. The total number of batches to be processed is the upper bound on this set.

The activity carried out by the operator in a sub-aisle $i$ for batch $b$ is described by the following parameters: $p_{bi}$ is the time needed to walk and collect the items, $l_{bi}$ is the time taken to load the picked items on AMR. The parameter $t_{ij}$ represents the time needed by the AMR to travel from one location $i \in V$ to location $j \in V$, while $u_b$ is the time needed to download the articles of a batch $b$ from the AMR at the end of the tour. Each customer order $o$ is associated with a due date $d_o$.

The coordination between the pickers and the AMR is regulated through the following variables: $a_{bi}$ denotes the moment the AMR handling batch $b$ arrives at sub-aisle $i$; $s_{bi}$ represents the moment the picker of sub-aisle $i$ starts collecting items of batch $b$; $h_{bi}$ is the moment the picker of sub-aisle $i$ starts loading the items of batch $b$ on the AMR. The completion time of each batch process by the AMR $a$ with position $k$ is denoted by $ct_{ak}$. The

assignment of each batch to a specific AMR tour is part of the optimization problem and is regulated by the binary indicator $x_{abk}$; it takes value 1 if batch $b$ is completed by AMR $a$ with position $k$; 0 otherwise. Starting from position $k = 0$ for each AMR, $k$ increases by one unit each time a batch is assigned to the AMR; the process ends when all batches have been assigned. For each sub-aisle $i$, the binary indicator $z_{bdi}$ takes values 1 if batch $b$ is processed before batch $d$ by the order picker; 0 otherwise. The tardiness of a customer order $o$ is denoted by $t_o$; the object of optimization is the minimization, for all orders, of the total tardiness. Note that the binary indicator $z_{bdi}$ only refers to a specific sub-aisle $i$ because, given two different warehouse sub-aisles $i \in V$ and $j \in V$ and two different batches $b \in B$ and $d \in B$, it can happen that batch $b \in B$ is processed before batch $d \in B$ by the picker of sub-aisle $i \in V$ and, on the contrary, batch $d \in B$ is processed before batch $\in B$ $b$ by the picker of sub-aisle $j \in V$.

For ease of reading, a summary of the main notation used for the models discussed along the paper can be found in Box 1.

$$ min \sum_{o \in O} t_o \tag{1} $$

$$ \sum_{b \in B} x_{abk} \leq 1 \; \forall a \in A; \; \forall k \in K \tag{2} $$

$$ \sum_{k \in K} \sum_{a \in a} x_{abk} = 1 \; \forall \, b \in B \tag{3} $$

$$ z_{ibd} + z_{idb} = 1 \; \forall \, b, d \in B; \; b \neq d; \; \forall i \in C_b; \; i \in C_d \tag{4} $$

$$ h_{bi} + l_{bi} - M \cdot (1 - z_{ibd}) \leq s_{di} \; \forall \, b, d \in B; \; b \neq d; \; \forall i \in C_b; \; i \in C_d \tag{5} $$

$$ s_{bi} + p_{bi} \leq h_{bi} \; \forall \, b \in B; \; \forall i \in C_b \tag{6} $$

$$ a_{bi} \leq h_{bi} \; \forall \, b \in B; \; \forall i \in C_b \tag{7} $$

$$ a_{bj} \geq h_{bi} + l_{bi} + t_{ij} \; \forall \, b \in B; \forall i \in C_b \backslash \{n+1\}; \; j \in C_b \backslash \{0\}; i < j \tag{8} $$

$$ ct_{ak} \geq ct_{ak-1} + h_{bi} + l_{bi} + u_b + t_{i,n+1} - M \cdot (1 - x_{abk}) \; \forall \, a \in A; \forall \, b \in B; \; \forall \, i \in C_b \{n+1\}; \; \forall k \in K \backslash \{1\} \tag{9} $$

$$ ct_{a1} \geq h_{bi} + l_{bi} + u_b + t_{i,n+1} - M \cdot (1 - x_{ab1}) \tag{10} $$

$$ a_{di} \geq ct_{ak} + t_{0,i} - M \cdot (2 - x_{abk} - x_{adk+1}) \; \forall \, a \in A; \; \forall \, k \in K; \; \forall \, b, d \in B; \; b \neq d; \; I \in Cd \tag{11} $$

$$ t_o \geq ct_{ak} - d_o - M \cdot (1 - v_{bo} \cdot x_{abk}) \; \forall \, a \in A; \; \forall \, k \in K; \; \forall \, o \in O; \; \forall \, b \in B \tag{12} $$

$$ a_{bi}, t_o, s_{bi}, ct_{ak}, h_{bi} \geq 0 \; \forall \, b \in B; \; \forall \, i \in V; \; \forall \, o \in O; \; \forall \, a \in A; \; \forall \, k \in K \tag{13} $$

$$ x_{abk}, z_{ibd} \in \{0,1\} \; \forall \, b, d \in B; \; \forall \, a \in A; \; \forall \, k \in K; \; \forall \, i \in V \tag{14} $$

**Box 1.** Main notation used along the paper

---

**Sets**
*B*: batches
*O*: customer orders
*A*: AMRs
*K*: sequencing numbers of a batch for each AMR
*V*: composed by the copies of the depot and the sub-aisles *(V = {0, 1, 2, ... , n, ... , 2n − 1, 2n, 2n + 1})*
$C_b$: relevant picking sub-aisles for batch $b \in B$
**Parameters**
$d_o$: due date of customer order $o \in O$
*M:* a sufficiently large positive number
$v_{bo}$: has value 1 if the customer order $o \in O$ is included in batch $b \in B$; 0 otherwise
$p_{bi}$: time the order picker requires to walk and retrieve the items of batch $b \in B$ which are stored in his picking sub-aisle $i \in V$
$l_{bi}$: time the order picker requires to pass the items of batch $b \in B$ stored in his picking sub-aisle $i \in V$ to the associated AMR
$u_b$: time required to unload the items of batch $b \in B$ from the associated AMR
$t_{ij}$: time an AMR requires to travel from the depot/handover location $i \in V$ to the depot/handover location $j \in V$
**Continuous decision variables**
$t_o$: tardiness of customer order $o \in O$
$a_{bi}$: arrival time of the AMR handling batch $b$ B at handover location of picking sub-aisle $i \in V$
$s_{bi}$: order picker's start time of picking the items of batch $b \in B$ stored in picking sub-aisle $i \in V$
$h_{bi}$: order picker's start time of passing the items of batch $b \in B$ stored in picking sub-aisle $i \in V$ to the associated AMR at the handover location
$ct_{ak}$: completion time of the batch completed by the AMR $a \in A$ with sequencing number $k \in K$
**Binary decision variables**
$x_{abk}$: takes value 1 if batch $b \in B$ is completed by AMR $a \in A$ with sequencing number $k \in K$; 0 otherwise
$z_{ibd}$: takes value 1 if batch $b \in B$ is handled before batch $d \in B$ by the order picker of subaisle $i \in V$; 0 otherwise
$q_{bi}$: takes value 1 if the items of batch $b \in B$ stored in sub-aisle $i \in V$ are collected in $i \in V$; 0 if if they are collected in $y(i) \in V$
**Function**
$y(i)$: function mapping each sub-aisle $i \in V/\{0, 2n + 1\}$ to the sub-aisle $y(i) \in V/\{0, 2n + 1\}$ that shares the same handover location

---

The objective of minimizing the total tardiness of all customer orders is defined in Constraint (1). Constraints (2) and (3) guarantee that the AMR processes the batches one at time. Constraints (4) define the order according to which the batches are to be processed by the order picker of each sub-aisle. Constraint (5) define the order picker's start time of picking the batch items stored in her/ picking sub-aisle. Constraints (6) and (7) guarantees that the order picker cannot hand over the picked items to the appropriate AMR until (i) she/he has returned to the handover location with these items (see Constraint (6)), and (ii) the AMR has arrived at her/his handover location (see Constraint (7)). Constraint (8) determine the arrival time of the AMR handling batch $b \in B$ at each handover location $j \in V$. Inequalities (9) guarantee that two batches do not overlap and ensure feasibility with respect to time. Inequalities (10) determine the completion time of the first batch of each AMR. Constraint (11) link the arrival time of the AMR handling batch $d \in B$ at the first relevant handover location $i \in C_d$ to the completion time of batch $b \in B$ (plus the time the AMR requires to travel from the depot to handover location $i \in C_d$) if the following holds: (i) batch $b \in B$ is handled by the same AMR as batch $d \in B$, and (ii) batch $b \in B$ is processed before batch $d \in B$ by the AMR. Constraint (12) calculate the tardiness for each customer order. Finally, the continuous decision variables and the binary decision variables are defined in Constraints (13) and (14), respectively.

The model presented can be algorithmically solved by black-box optimizers in short times (see Section 5). Once the optimal assignment of the variables of the model is found, the optimal operational scheduling for the pickers can be derived as follows. The sequence of picking actions requested to the worker operating in sub-aisle *i* are obtained from the values of the timing variables $s_{bi}$, that contain for each batch *b* the (eventual) start of the picking process. It is straightforward to derive a user-friendly list for each picker starting from this information. Combining the optimal value of the set of variables *x* and *ct* it is finally possible to derive the sequence of operations carried out by the AMRs and to consequently program them.

### 4.2. Separated Handover Locations for Sub-Aisles

In the second implementation considered, and presented below, the same handover location is shared between the two pickers operating in a pair of sub-aisles, as depicted in Figure 3. The AMR can visit each handover location twice, either while moving forward or backward in the corridor. As in the implementation described above in Section 4.1, the AMR can eventually turn back to the depot when the right-most aisle containing an order is reached.

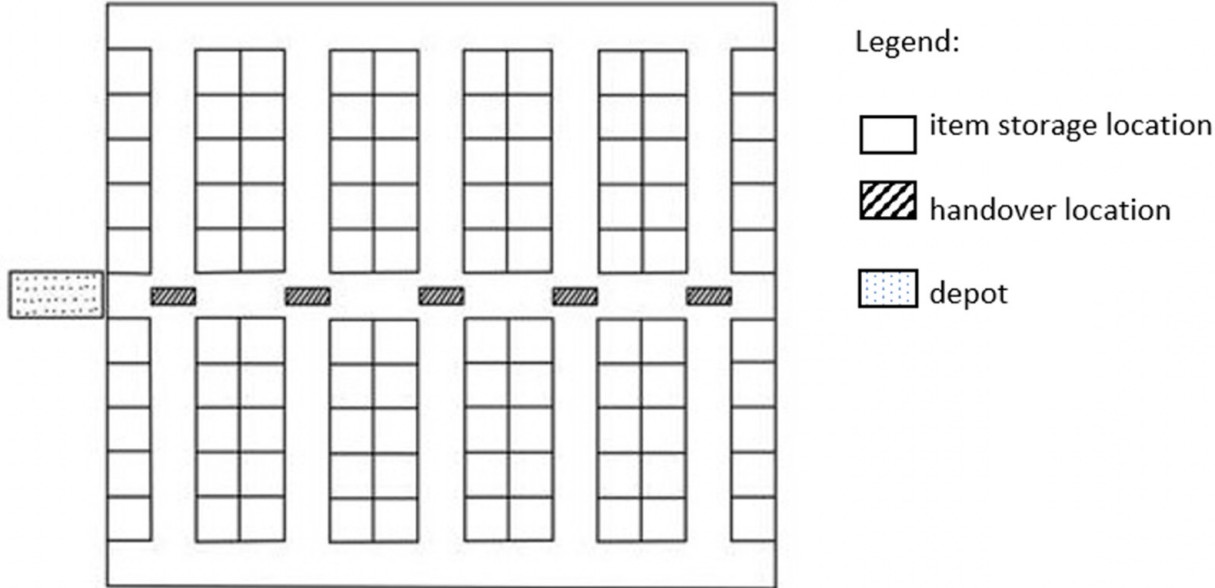

**Figure 3.** Warehouse with a two-blocks layout and shared handover locations.

A pair of opposite sub-aisles is associated with the same handover location. There is a total of *n* pairs (*n* + 1 pairs, if the one consisting of the depot instances is also considered). The special feature of this configuration is that the picker of each sub-aisle has the option of loading the AMR either when it moves towards the right-most corridor, or in its way back towards the depot. This choice is possible because, as previously anticipated, according to this warehouse structure, the pair of aisles share the same handover location.

Note that this implementation adds complexity and requires more synchronization than the one discussed in Section 4.1, but provides greater flexibility, both in the coordination between different operators and in the coordination between the operator and the cart. This will hopefully lead to more efficient solutions.

### A Mixed Integer Linear Programming Formulation

Concerning the mathematical model formulation of this second two-block warehouse configuration, the differences from model described in Section 4.1.1 are highlighted below. The numbering of the sub-corridors is done clockwise. The set *Y* contains the pairs of sub-corridors that are associated with the same handover location and are denoted by *i*

and $y(i)$, respectively. For each corridor $i$ belonging to the set $V$, the associated corridor $y(i)$ in $Y$ in the pair is calculated as:

$$y(i) = 2n - i + 1$$

where $n$ is the number of sub-corridors per block and, consequently, handover locations. This formula is also valid for depot instances $0$ and $2n + 1$. Finally, the binary variable $q_{bi}$ takes different values, depending on whether the operator of a corridor $i$ loads the items on the AMR in correspondence to the visit for its corridor or in that associated to corridor $y(i)$.

For the sake of clarity, we present the model in its integrity, although some of the constraints had been preciously introduced already. We also remind the reader that a summary of the notation used can be found in Box 1.

$$min \sum_{o \in O} t_o \tag{15}$$

$$\sum_{b \in B} x_{abk} \leq 1 \ \forall a \in A; \ \forall k \in K \tag{16}$$

$$\sum_{k \in K} \sum_{a \in a} x_{abk} = 1 \ \forall b \in B \tag{17}$$

$$z_{ibd} + z_{idb} = 1 \ \forall b, d \in B; \ b \neq d; \ \forall i \in C_b; \ i \in C_d \tag{18}$$

$$h_{bi} + l_{bi} - M \cdot (1 - z_{ibd}) \leq s_{di} \ \forall b, d \in B; \ b \neq d; \ \forall i \in Cb; i \in Cd \tag{19}$$

$$q_{bi} = 1 \ \forall b \in B; \ i = 0; \ i = 2n + 1 \tag{20}$$

$$s_{bi} + p_{bi} \leq h_{bi} \ \forall b \in B; \ \forall i \in Cb \tag{21}$$

$$h_{bi} \geq a_{bi} - M \cdot (1 - q_{bi}) \ \forall b \in B; \ \forall i \in Cb \tag{22}$$

$$h_{bi} \geq a_{by(i)} - M \cdot q_{bi} \ \forall b \in B; \ \forall i \in Cb \tag{23}$$

$$a_{bj} \geq h_{bi} + l_{bi} + t_{ij} - M \cdot \left(2 - q_{bi} - q_{bj}\right) \ \forall b \in B; \ \forall I \in Cb \backslash \{2n+1\}; \ j \in Cb \backslash \{0\}; \ j > i \tag{24}$$

$$a_{bj} \geq h_{bi} + l_{bi} + t_{y(i)j} - M \cdot q_{bi} - M \cdot \left(1 - q_{bj}\right) \ \forall b \in B; \ \forall i \in Cb \backslash \{2n+1\}; \ j \in Cb \backslash \{0\}; \ j > y(i) \tag{25}$$

$$a_{by(j)} \geq h_{bi} + l_{bi} + t_{iy(j)} - M \cdot q_{bj} - M \cdot (1 - q_{bi}) \ \forall b \in B; \ \forall i \in Cb \backslash \{2n+1\}; \ j \in Cb \backslash \{0\}; \ y(j) > i \tag{26}$$

$$a_{by(j)} \geq h_{bi} + l_{bi} + t_{y(i)y(j)} - M \cdot q_{bj} - M \cdot q_{bi} \ \forall b \in B; \ \forall i \in Cb \backslash \{2n+1\}; \ j \in Cb \backslash \{0\}; \ y(j) > y(i) \tag{27}$$

$$ct_{ak} \geq ct_{ak-1} + h_{bi} + l_{bi} + u_b + t_{i,2n+1} - M \cdot (2 - x_{abk} - q_{bi}) \ \forall a \in A; \ \forall b \in B; \ \forall k \in K \backslash \{1\}; \ i \in Cb \tag{28}$$

$$ct_{ak} \geq ct_{ak-1} + h_{bi} + l_{bi} + u_b + t_{y(i),2n+1} - M \cdot (1 - x_{abk}) - M \cdot q_{bi} \ \forall a \in A; \ \forall b \in B; \ \forall k \in K \backslash \{1\}; \ i \in Cb \tag{29}$$

$$ct_{a1} \geq h_{bi} + l_{bi} + u_b + t_{i,2n+1} - M \cdot (2 - x_{ab1} - q_{bi}) \ \forall a \in A; \ \forall b \in B; \ i \in Cb \tag{30}$$

$$ct_{a1} \geq h_{bi} + l_{bi} + u_b + t_{y(i),2n+1} - M \cdot (1 - x_{ab1}) - M \cdot q_{bi} \ \forall a \in A; \ \forall b \in B; \ \forall i \in Cb \tag{31}$$

$$a_{di} \geq ct_{ak} + t_{0,i} - M \cdot (2 - x_{abk} - x_{adk+1}) \ \forall a \in A; \ \forall k \in K; \ \forall b, d \in B; \ b \neq d; \ i \in Cd \tag{32}$$

$$t_o \geq ct_{ak} - d_o - M \cdot (1 - v_{bo} \cdot x_{abk}) \ \forall a \in A; \ \forall o \in O; \ \forall k \in K; \ \forall b \in B \tag{33}$$

$$a_{bi}, t_o, s_{bi}, ct_{ak}, \ h_{bi} \ \geq 0 \forall b \ \in \ B; \ \forall I \ \in \ V; \ \forall o \ \in \ O; \ \forall a \ \in \ A; \ \forall k \ \in \ K \tag{34}$$

$$x_{abk}, z_{ibd}, q_{bi} \in \{0,1\} \ \forall b, \ d \ \in \ B; \ \forall a \ \in \ A; \ \forall k \ \in \ K; \ \forall i \ \in \ V \tag{35}$$

The objective of minimizing the total tardiness of all customer orders is defined in Constraint (15). Constraints (16) and (17) guarantee that the AMR processes the batches one at time. Constraint (18) define the order according to which the batches are to be processed by order picker of sub-aisle $i \in V$. Constraint (19) define the order picker's start time of picking the batch items stored in her/his picking sub-aisle. Constraint (20) guarantee that the AMR must start and finish its travel in the depot. Constraints (21)–(23) guarantees that the order picker cannot hand over the picked items to the appropriate AMR until (i) she/he has returned to the handover location with these items (see Constraint (21)), and (ii) the AMR has arrived at the associated handover location (see Constraints (22) and (23)). Constraints (24) and (25) determine the arrival time of the AMR handling batch $b \in B$ at each sub-aisle $j \in C_b$ when items of sub-aisle $i$ are picked in $i$ (see Constraint (24)) and when the items of $i$ are picked in $y(i)$ (see Constraint (25)). Constraints (26) and (27) determine the arrival time of the AMR handling batch $b \in B$ at each couple of sub-aisle $y(j)$ of $j \in C_b$ when items of sub-aisle $i$ are picked in $i$ (see Constraint (26)) and when the items of $i$ are picked in $y(i)$ (see Constraint (27)). Inequalities (28) and (29) determine the completion time of each batch processed by the AMR $a \in A$ with sequencing number $k \in K$. Inequalities (30) and (31) determine the completion time of the first batch of each AMR. Constraint (32) link the arrival time of the AMR handling batch $d \in B$ at the first relevant handover location $i \in C_d$ to the completion time of batch $b \in B$ (plus the time the AMR requires to travel from the depot to handover location $i \in C_d$) if the following holds: (i) batch $b \in B$ is handled by the same AMR as batch $d \in B$, and (ii) batch $b \in B$ is processed before batch $d \in B$ by the AMR. Constraint (33) calculate the tardiness for each customer order. Finally, the continuous decision variables and the binary decision variables are defined in Constraints (34) and (35), respectively.

The logic for deriving the picking lists for the workers associated with the optimal solution of the mathematical model is analogous to that discussed in Section 4.1.1.

## 5. Numerical Experiments

The mathematical formulations presented in Section 4 were implemented in Python and solved using the black-box solver Gurobi (https://www.gurobi.com/ accessed on 4 November 2022) 9.5.1. All the experiments were run on a computer equipped with an AMD Ryzen 5 3500U processor and 8 GB of RAM, and all the computation times were negligible.

In the study we present, the system is simulated for three instances of different sizes consisting of 50, 150, and 300 items, respectively. For each item, the identifying number (Article Number), the order to which it belongs (Customer Order), the time the order should ideally be ready for dispatching (Due Date), the aisle in which it is stored, the specific aisle rack (House), and the number of units required (Number of packages in SKU) are known.

### 5.1. Details of the Case Study

Figures 4 and 5 show the distances of the two-blocks layout with separated and shared handover locations, respectively.

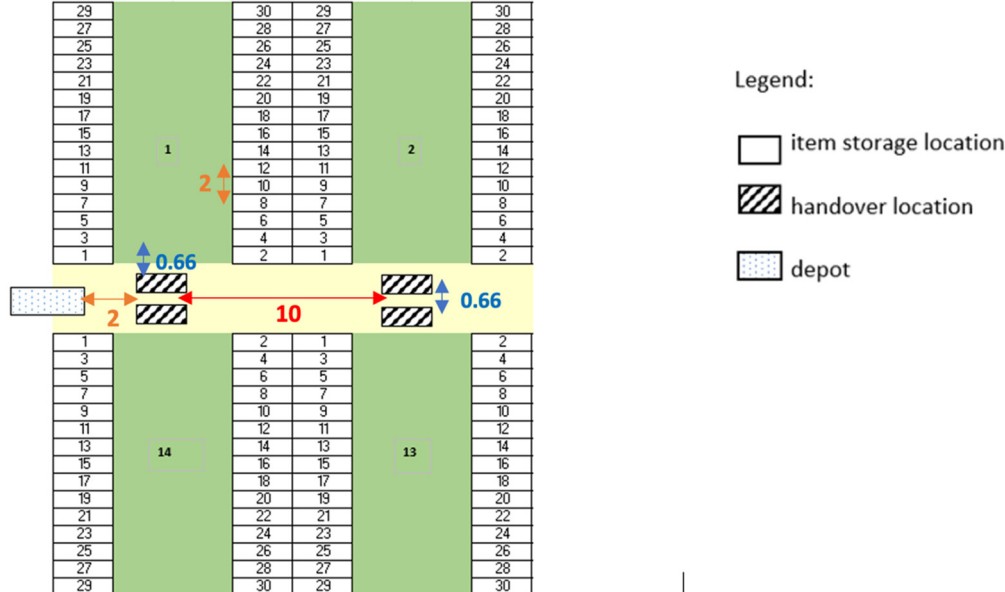

**Figure 4.** Distances in the two-blocks layout with separated handover locations (meters).

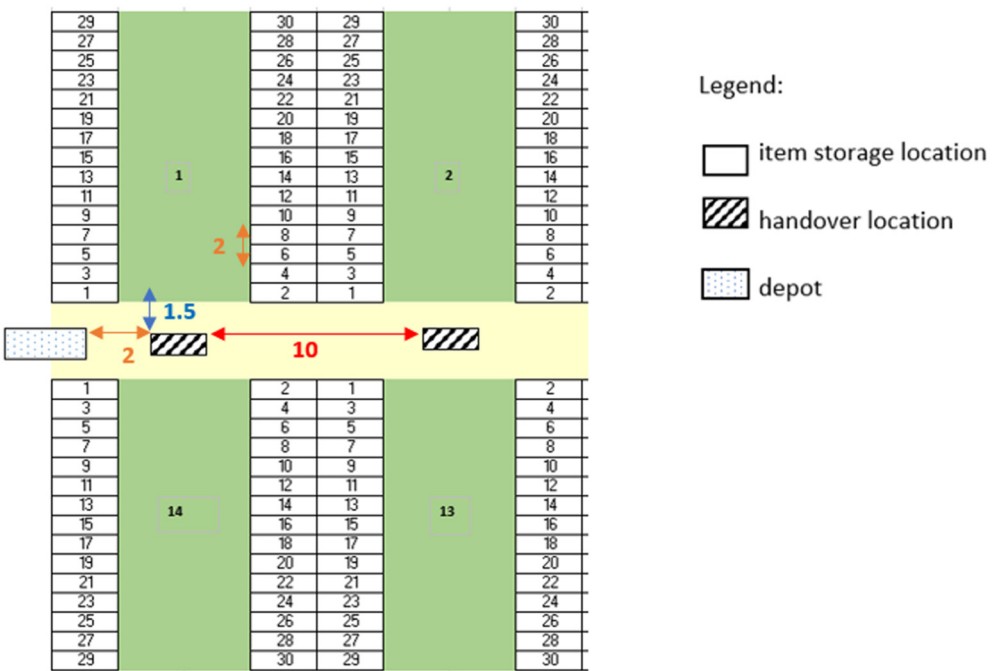

**Figure 5.** Distances in the two-blocks layout with shared handover locations (meters).

In both the configurations, each sub-corridor consists of 30 houses, 15 on the left and 15 on the right. As can be seen in the figures, the houses are numbered from 1 to 30, starting with those closest to the handover location associated with the aisle. The warehouse consists of 7 handover locations and 14 sub-aisles, which are numbered clockwise. The dimensions and parameters of that warehouse, as well as the standard operator and AMR activity times, are as follows, and represent an instance of the general layout depicted in Figure 4:

- Distance between two close handover locations (meters): 10
- Distance between the depot and the leftmost handover location (meters): 2
- Distance between two adjacent storage locations (houses) (meters): 2

- Distance between each handover location and the first storage location of the corridor (meters): 3

The picking strategy used by the aisle operator, it is the same as that proposed in [21]: the picker, starts from the handover location of his aisle, begins by picking the products stored in the farthest house among those containing items of the lot he has been assigned, and then continues picking moving towards the nearest products in his return route, always picking from both the right and left houses.

The following data are related to operator and AMR activity time and speed:

- Unit picking time (seconds): 8
- Unitary loading/unloading time (time required to load/unload a unit from the cart, seconds): 4
- Walking time of the operator between two adjacent storage locations (seconds): 3
- Traveling time of the AMR between two consecutive handover locations (seconds): 15
- Traveling time of the AMR from the storage location to the first handover location (seconds): 3
- Traveling time of the operator from the handover location to the first storage location (seconds): 4.5

We neglect for simplicity (and consistently with [21]) the time taken by the operator to walk from the handover location in his aisle to the first storage location. The distance between the handover location and the first house of each sub-corridor measures, 0.66 m and 1.5 m, for the separated and shared locations.

### 5.2. Experiments on the AMS Fleet Size and Speed

This section analyzes the impact that increasing cart speed and fleet size have on total tardiness. Indeed, these are parameters that can be easily changed, with a related technological intervention, without the need to reorganize the entire picking system [21].

Two different fleet sizes are considered, consisting of 2 and 3 carts, respectively; with larger fleet sizes, the tardiness goes consistently to zero for each instance considered, and thus the results are omitted. For AMR speed, expressed in km/h, we consider the following values: {2.4, 4.6, 4.8, 6.0, 7.2}.

As anticipated in Section 4.1, the experiments are carried out for three instances of different sizes, consisting of 50 (SMALL), 150 (MEDIUM) and 300 items (LARGE), respectively. The impact of AMR speed on total tardiness can be observed for order instances of three different sizes. For each group of instances, SMALL, MEDIUM, and LARGE, respectively, and for each possible combination of the number of AMRs and speed, the corresponding tardiness values are shown respectively in Tables 1 and 2 (for the cases with separated and shared handover locations, respectively).

**Table 1.** Values of tardiness as a function of the AMR speed—separated handover locations.

| | Small | Medium | | Large | |
|---|---|---|---|---|---|
| AMR Speed (km/h) | Number of AMR = 2 | Number of AMR = 2 | Number of AMR = 2 | Number of AMR = 2 | Number of AMR = 3 |
| 2.4 | 2634 | 5645 | 4131 | 16,954 | 1870 |
| 3.6 | 0 | 4863 | 2312 | 14,820 | 0 |
| 4.8 | 0 | 3559 | 2241 | 13,173 | 0 |
| 6.0 | 0 | 3513 | 2227 | 12,869 | 0 |
| 7.2 | 0 | 2953 | 2112 | 11,314 | 0 |

**Table 2.** Values of tardiness as a function of the AMR speed—shared handover locations.

| | Small | Medium | | Large | |
| --- | --- | --- | --- | --- | --- |
| AMR Speed (km/h) | Number of AMR = 2 | Number of AMR = 2 | Number of AMR = 2 | Number of AMR = 2 | Number of AMR = 3 |
| 2.4 | 1269 | 7341 | 2945 | 10,207 | 95 |
| 3.6 | 0 | 2981 | 2085 | 9885 | 0 |
| 4.8 | 0 | 2946 | 1656 | 9912 | 0 |
| 6.0 | 0 | 2857 | 1245 | 8603 | 0 |
| 7.2 | 0 | 1250 | 904 | 3199 | 0 |

As it can be seen from Tables 1 and 2, an increase in AMR speed, has a positive impact on tardiness. For example, a relatively small increase in the AMR speed from 2.4 km/h to 3.6 km/h for a medium-sized instance leads to remarkable decreases in tardiness. It can be seen, however, that in other cases, by increasing the speed, the decrease in tardiness is very marginal. This is due to the pickers' picking time, that becomes the bottlenecks of the system; although the cart takes less time to reach the handover location, it must wait for the picker, who has not yet finished picking.

Given a constant AMR speed, increasing fleet size from 2 to 3 can have a significant impact on tardiness. An increase 2 to 3 of the number of AMR leads to a crucial decrease in tardiness for large instances. In general, the larger the number of orders, the greater the impact of increased fleet and speed on total tardiness. The presence of bottlenecks represented by human pickers can be observed also in these experiments.

The experiments showed that an increase in these factors leads to an improvement in total tardiness, consistent with what was obtained in [21]. However, analysis of the results shows that the percentage improvement in tardiness after increasing fleet size is greater than the one obtained by the adoption of faster AMRs. This conclusion disagrees with what was stated in [21]. It can be argued that this result can be blamed on the size of the picking area, which is large compared to the number of aisles, and by the fact that the two-blocks layout creates different situations.

Comparing the two implementations of the system, it appears that with shared handover location, the maximum percentage decrease in tardiness following an increase in fleet size. This configuration, moreover, turns out to be more efficient from the point of view of bottlenecks imputed to operators; the presence of a shared handover location, in fact, ensures greater coordination between operator and truck, limiting the latter's waiting time.

In general, a possible solution to limit bottlenecks could be to employ two operators per aisle, consequently speeding up the overall picking process.

*5.3. Comparison of the Implementations*

From the analyses of the results obtained in the previous section, it is possible to compare the different warehouse layouts and derive the statements presented below.

5.3.1. Tardiness Values for Different Instance Sizes

The values of tardiness as a function of instance size are given for each warehouse layout; the graphical representation is shown in Figure 6. These values are derived for a fleet size of 2 and an AMR speed of 2.4 km/h.

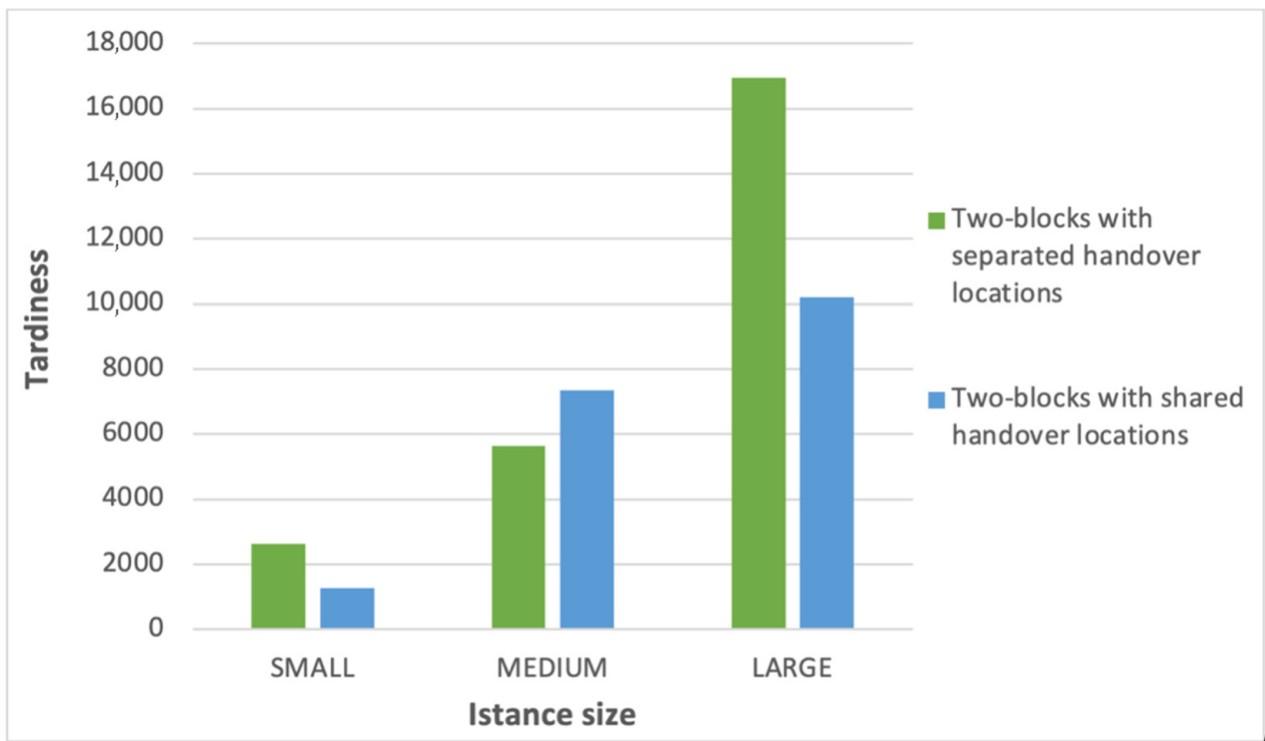

**Figure 6.** Tardiness values for different layouts and instance sizes.

From the results presented in Figure 6, it is remarkable that how for medium size instances the implementation with separated handover locations emerges as the most efficient one; in the other two cases of small and large instance sizes, the one with shared handover locations is optimal instead. This happens because, in the case of a small instance, the complexity of the problem is low and the layout with shared handover locations allows minimizing the waiting time of both picker and AMR, without excessive coordination effort between complementary sub-corridors. The result achieved for the medium-sized instance is more meaningful. In that case, the layout with separated location is optimal since it is more intuitive for a limited number of items in each sub-corridor while the alternative layout probably only adds complexity without providing any improvement to tardiness. Finally, for a large instance, the ability to load items in the complementary aisle leads to a significant improvement in tardiness. Therefore, the layout with shared locations is indeed optimal: in the presence of many items per sub-corridor it significantly decreases the waiting time of both the picker and the AMR, improving coordination between the two.

5.3.2. Tardiness Values for Different AMR Fleet Sizes and Speeds

The tardiness results obtained by increasing the number of AMRs from 2 to 3 and increasing the AMR speed from 2.4 to 3.6 km/h are shown below. These values are illustrated in Figures 7 and 8, respectively.

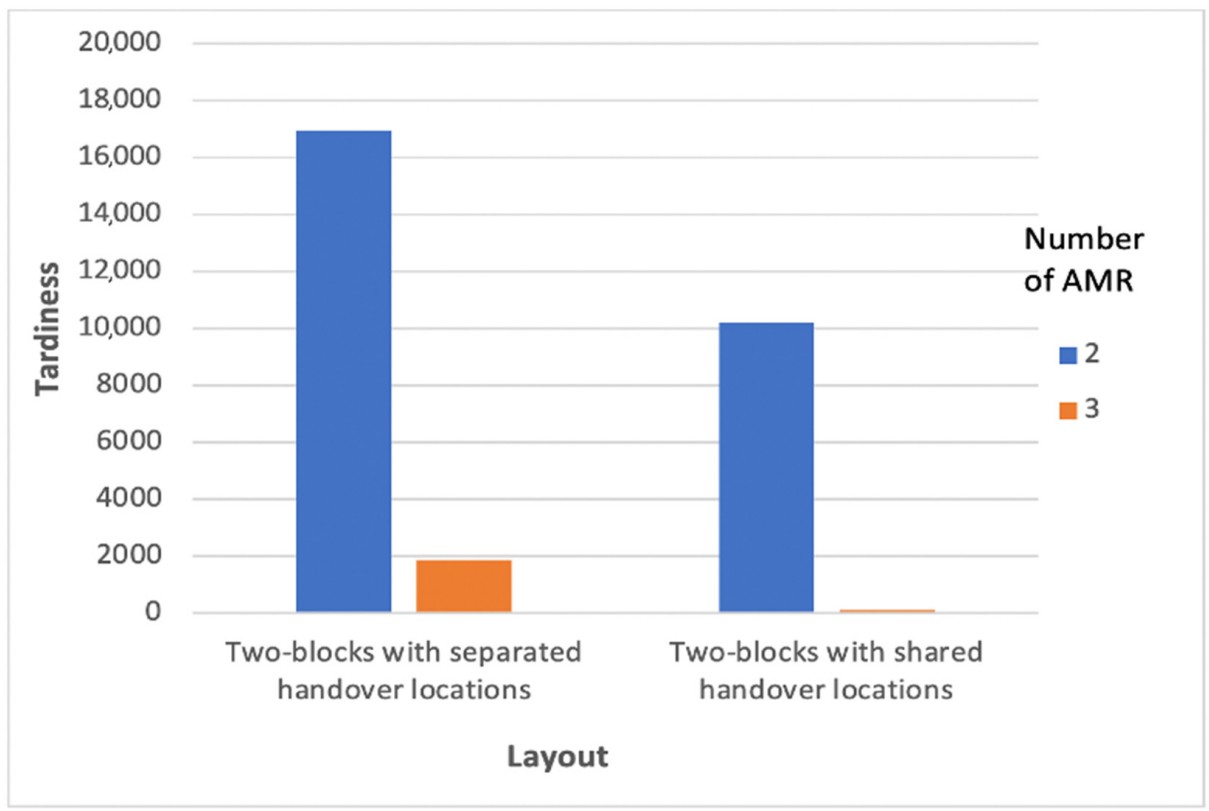

**Figure 7.** Tardiness values for increasing fleet sizes.

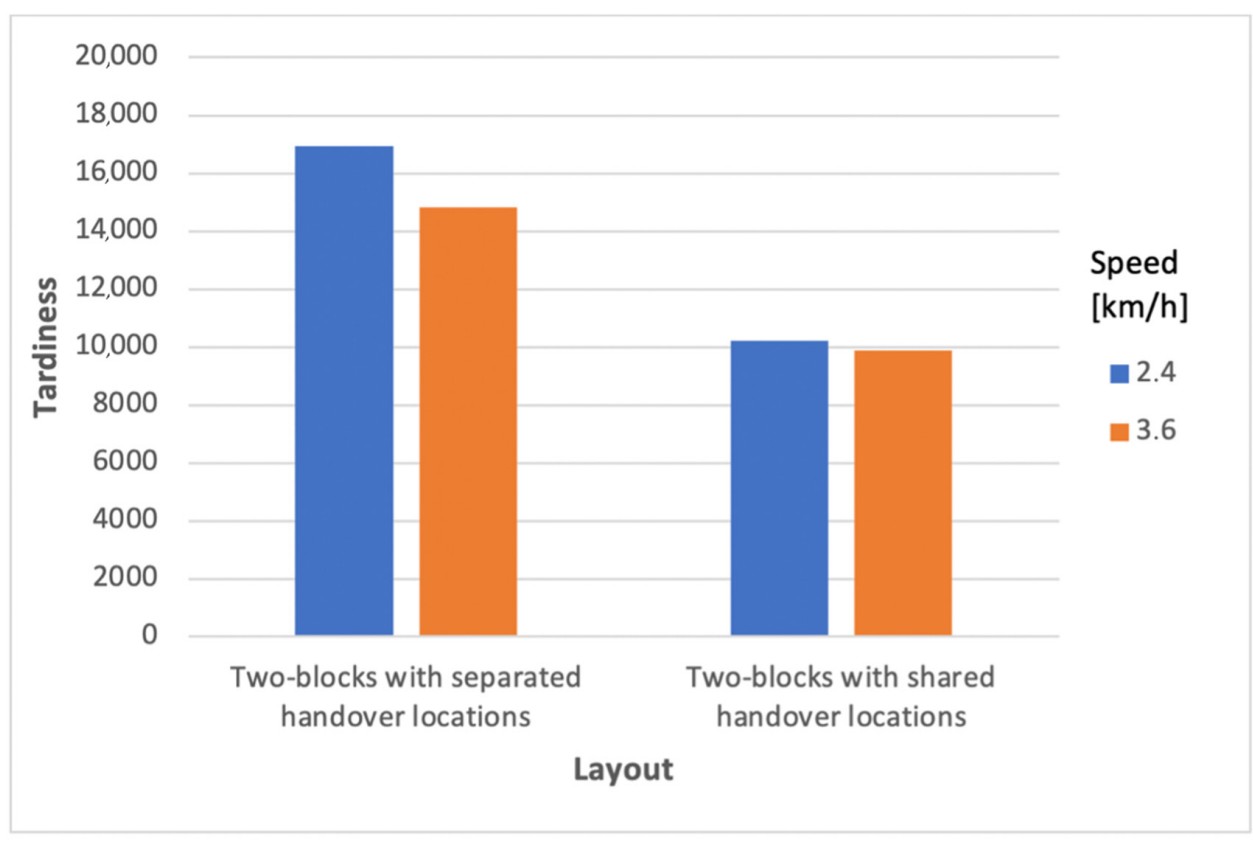

**Figure 8.** Tardiness values for increasing AMR speeds.

The previous values refer to a large instance. The analysis on the AMR fleet was carried out considering a trolley speed of 2.4 km/h; however, the tardiness analysis was carried out on a fleet consisting of 2 trolleys. As can be seen from Figures 7 and 8, the most efficient layout, from the point of view of total tardiness, turns out to be the one consisting of two blocks and shared handover location, before and after the interventions on AMR fleet and speed.

### 5.3.3. Tardiness Percentage Improvements for Different Implementations and Conditions

Then, the percentage improvement in tardiness connected to the previous increases in fleet size and trolley speed, for the three different warehouse layouts, is shown in in Figure 9.

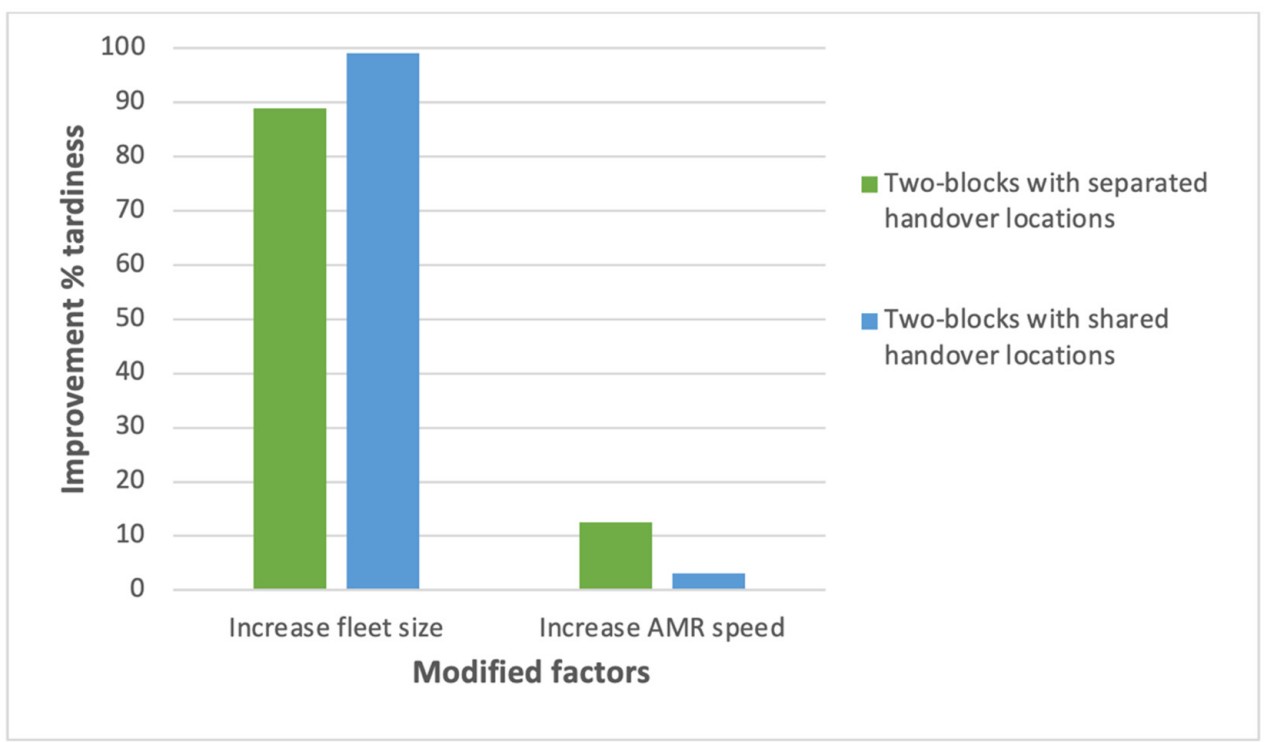

**Figure 9.** Tardiness improvement for different solutions.

As can be seen, the maximum percentage improvement in tardiness after increasing fleet size is found in the two-block layout with shared handover location (Figure 3). In contrast, the maximum percentage improvement in tardiness after increasing AMR speed can be found in the two-block layout with separated handover location (Figure 2). However, for each of the three warehouse layouts presented, it is interesting to note that the percentage improvement in tardiness after having increased the fleet size is much greater than that achieved by increasing AMR speed.

For all the warehouse configurations presented, moreover, the larger the instance size, the greater the percentage improvement in tardiness after increasing both factors.

In conclusion, it can be said that the layout in which each handover location is shared by two sub-aisles (Figure 3) achieves maximum efficiency with large instances and with limited speed and number of AMRs (Figure 6). In addition, it is concluded that an increase in the size of the AMR fleet, rather than an increase in AMR speed, has a much greater impact on total tardiness; this turns out, therefore, to be the most strategical choice for the company.

## 6. Conclusions

The present study analyzes a warehouse with a two-block layout and proposes optimization models for two possible implementations of an AMR-assisted hybrid order picking system. The first solution is standard, with separated handover locations for each sub-aisle, while the second allows the pickers operating on pairs of sub-aisles facing each other to share the handover location. This increases the complexity of the optimization but potentially leads to more efficient solutions in terms of total tardiness. The optimization is carried out by attacking two novel mathematical programming models we propose via black-box solvers.

The study is conducted on real-world data originating from a German brick-and-mortar grocery retailer and the experimental simulations carried out suggest that a layout consisting of two blocks and separated handover locations is more efficient for a limited number of items; as the size of the instance increases, however, the layout consisting of two blocks and shared handover location is more optimal. This happens because, for small order instances, the high cart loading times in the second layout mentioned are associated with sub-corridors with very low quantities of items to be picked. As instance size increases, on the other hand, the higher loading times in that configuration are offset by lower cart waiting times in each sub-corridor, due to the greater flexibility of the layout, and higher quantities of items to be picked in each sub-corridor.

It is possible to conclude, therefore, that for a company handling many items, the most efficient implementation for an AMR system in a two-block warehouse is the one making use of shared handover location; this configuration, compared to the other one, also allows for the greatest percentage improvement in tardiness following the increase in fleet size. Increasing the number of AMRs turns out to be the most strategically cost-effective intervention.

In addition, experiments conducted on fleet size and cart speed showed that an increase in these factors leads to an improvement in total tardiness, consistent with what observed in the previous literature. However, the analysis of the results shows that the percentage improvement in tardiness after increasing fleet size is greater than the one obtained by increasing speed of AMRs. The conclusion disagrees with some of the results previously published. This happens because of the two-blocks layout configuration and because the size of the picking area is large compared to the number of aisles in the real warehouse considered and, therefore, the work of the human pickers plays an important role in determining the total tardiness.

**Author Contributions:** Conceptualization, G.P. and R.M.; methodology, G.P. and R.M.; software, G.P. and X.C.; validation, G.P.; formal analysis, G.P. and R.M..; investigation, G.P. and X.C.; resources, D.L. and M.K.; data curation, G.P., D.L. and M.K..; writing—original draft preparation, G.P. and R.M.; writing—review and editing, G.P., X.C., D.L., M.K. and R.M.; visualization, G.P.; supervision, R.M.; project administration, R.M. All authors have read and agreed to the published version of the manuscript.

**Funding:** This research received no external funding.

**Data Availability Statement:** The data presented in this study are available on request from the corresponding author. The data are not publicly available due to their confidential nature.

**Conflicts of Interest:** The authors declare no conflict of interest.

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
