# Peer review of "AMR-Assisted Order Picking: Models for Picker-to-Parts Systems in a Two-Blocks Warehouse"

_algorithms, doi:10.3390/a15110413_

Round 1
Reviewer 1 Report
As a result of the review, I have come to the conclusion that the manuscript requires major revision.
It is the purpose of this manuscript to develop a mathematical programming model to generate picking orders for picker-to-parts systems in a two-block warehouse. It is well written and organized. However, the engineering section (section 4, line 176 of page 5) is poorly presented. Before explaining the details of a MIP model, the indexes, sets, and (decision) variables are usually provided.
I am unsure whether the formulas of the proposed model are current? For example, an MIP model would not distinguish between Zibd and Zibb, as shown in equation (4) on page 6 (line 237). In the case of Z123, does it refer to Zibd or Zidb? Therefore, I am wondering whether the proposed MIP model can be implemented? It is suggested that the manuscript include the Python code for the MIP model.
In addition to the performance comparisons of the scalability of the proposed MIP model, the interpretation of decision variables should also be provided. Is it possible to interpret the optimal solution in terms of its decision variables? What are the order-picking lists formed by those decision variables? Consequently, the MIP model may not contribute as much to order picking in warehouses without this feature.
Author Response
C:
As a result of the review, I have come to the conclusion that the manuscript requires major revision.
It is the purpose of this manuscript to develop a mathematical programming model to generate picking orders for picker-to-parts systems in a two-block warehouse. It is well written and organized. However, the engineering section (section 4, line 176 of page 5) is poorly presented. Before explaining the details of a MIP model, the indexes, sets, and (decision) variables are usually provided.
A:
On top of the description of the notation within the text, we have now added Table 1 that summarises the notation used along the paper. Moreover, a description of how the results of the optimization can be engineered back into operational concepts has been added around line 280 (and 373). Some other minor improvements have been done to the section.
C:
I am unsure whether the formulas of the proposed model are current? For example, an MIP model would not distinguish between Zibd and Zibb, as shown in equation (4) on page 6 (line 237). In the case of Z123, does it refer to Zibd or Zidb? Therefore, I am wondering whether the proposed MIP model can be implemented? It is suggested that the manuscript include the Python code for the MIP model.
A:
The formulae are correct, and follow the standard semantic used for describing MIPs. This is corroborated by our Python implementation that provides sound operational assignments. We however do not share our implementation at this stage, and the reason is twofold: the work is a collaboration with a Company and the code is still a research asset for us.
C:
In addition to the performance comparisons of the scalability of the proposed MIP model, the interpretation of decision variables should also be provided. Is it possible to interpret the optimal solution in terms of its decision variables? What are the order-picking lists formed by those decision variables? Consequently, the MIP model may not contribute as much to order picking in warehouses without this feature.
A: The interpretation of the decision variables was not obvious, so we added a detailed description at line 280. Some reword in the explanations about the difference of the assignments in the two scenarios have also been done.
Reviewer 2 Report
In this manuscript the authors present an application of a hybrid picker-to-parts order 17 picking system, in which human operators collaborate with Automated Mobile Robots (AMRs). Although the topic of the manuscript is interesting and worthy of investigation, there are several issues that should be resolved by the authors. 1. Please revise the Abstract Section in order to eliminate common knowledge and highlight in a more clear way the contribution of the presented research work. 2. The Literature Review Section should be further elaborated in order to explicitly discuss what has been done in similar research works highlighting the limitations and challenges, and how the presented manuscript contributes to the field. 3. It is recommended to reduce the mathematical equations to the bare minimum required. 4. The authors improve numerical results presentation for system. 5. Please make sure that the quality of the figures is acceptable. Concretely 300dpi is the threshold resolution. 6. The completeness of the literature review should be further elaborated with the addition of more recent and relevant publications such as:
-
An innovative model to optimise inventory management: A case study in healthcare sector
Forcina, A., Petrillo, A., Di Bona, G., De Felice, F., Silvestri, A. ,
Author Response
C:
In this manuscript the authors present an application of a hybrid picker-to-parts order 17 picking system, in which human operators collaborate with Automated Mobile Robots (AMRs). Although the topic of the manuscript is interesting and worthy of investigation, there are several issues that should be resolved by the authors.
1. Please revise the Abstract Section in order to eliminate common knowledge and highlight in a more clear way the contribution of the presented research work.
A: The abstract has been reworded and expanded as indicated.
C:
2. The Literature Review Section should be further elaborated in order to explicitly discuss what has been done in similar research works highlighting the limitations and challenges, and how the presented manuscript contributes to the field.
A: The Literature Review Section has been expanded with more discussion and especially more recent references, as suggested also by another reviewer.
C:
3. It is recommended to reduce the mathematical equations to the bare minimum required.
A: The models are an important contribution of the work, so we preferred to leave them written as clear as possible. We however added Table 1 with a summary of the notation used, in order help the reader. This was suggested also by the other reviewers.
C:
4. The authors improve numerical results presentation for system.
A: The results section has been revised, while at line 280 a clear explanation on how assignment are obtained from the results of the optimization has been added.
C:
5. Please make sure that the quality of the figures is acceptable. Concretely 300dpi is the threshold resolution.
A: All the figures respect the thresholds of the Journal.
C:
6. The completeness of the literature review should be further elaborated with the addition of more recent and relevant publications such as:
An innovative model to optimise inventory
management: A case study in healthcare sector Forcina, A., Petrillo, A., Di Bona, G., De Felice, F., Silvestri, A. International Journal of Services and Operations Management, 2017, 27(4), pp. 549–568
A: The paper has been added together with several other new ones.
Reviewer 3 Report
This paper a warehouse with two-blocks layout is investigated and two alternative implementations for an AMR system are considered.
The authors may need to take into consideration the following issues:
1. The abstract section is a little bit short. I suggest the authors can add more paragraph to mention about contributions.
2. References only have 1 reference have 3-years (since 2020) publications. Thus, I suggest the author may add more description about the state-of-the-art techniques and to cite some top journals of the article.
3. There are too many notations in section 4. I suggest the authors can collect all notations into one Table.
4. I suggest the authors can provide a pseudo-code or flowchart to mention their proposed model.
5. I suggest the authors should compare the proposed model with the SOTA model.
Author Response
C:
This paper a warehouse with two-blocks layout is investigated and two alternative implementations for an AMR system are considered.
The authors may need to take into consideration the following issues:
1. The abstract section is a little bit short. I suggest the authors can add more paragraph to mention about contributions.
A: The abstract has been extended as requested.
C:
2. References only have 1 reference have 3-years (since 2020) publications. Thus, I suggest the author may add more description about the state-of-the-art techniques and to cite some top journals of the article.
A: Five references from 2021/22 have been added and the discussion slightly revised.
C:
3. There are too many notations in section 4. I suggest the authors can collect all notations into one Table.
A: Table 1 now contains a summary of the notation used along the paper.
C:
4. I suggest the authors can provide a pseudo-code or flowchart to mention their proposed model.
A: The model is attacked directly by a black-box solver, and we make this clear in the text now. We also added an explanation at line 280 on how to translate back the results of the optimization into operational assignments. We believe this moves along the line suggested by the reviewer although it is not a direct implementation of the suggestion.
C:
5. I suggest the authors should compare the proposed model with the SOTA model.
A: We adapted a SOTA model to our specific layout, and as far as we are aware this is the first study for a two-block layout (although the configuration is fairly common in warehouses). It is therefore not possible to have a direct comparison with other models.
Round 2
Reviewer 1 Report
Authors are responsible for providing proof of approach and performance validation. Nonetheless, I do not believe the responses from authors who simply state that the model they provide is correct and that the decision variables are not obvious. Are the authors unable to interpret their variables? Therefore, I believe the Python code, or at least the pseudo-code, is required in order to demonstrate the accuracy of their model.
Author Response
Q:
Authors are responsible for providing proof of approach and performance validation. Nonetheless, I do not believe the responses from authors who simply state that the model they provide is correct and that the decision variables are not obvious.
A: The model is properly described and in our opinion correct. We do not know what to add to this statement. If any problem can be spot, we are happy to fix it.
In the previous review round the comment was: "I am unsure whether the formulas of the proposed model are current? For example, an MIP model would not distinguish between Zibd and Zibb, as shown in equation (4) on page 6 (line 237). In the case of Z123, does it refer to Zibd or Zidb? Therefore, I am wondering whether the proposed MIP model can be implemented?". There is not a punctual critic, just general speculations the model might be incorrect. The comment on Z123 was the only one we could address directly and explained that MILPs work like this.
Q:
Are the authors unable to interpret their variables?
A: We give a clear explanation of every single variable, so I believe we know what we are speaking about. As explained in our previous answer, now we also provide a clear explanation about the interpretation of the variables, that was indeed missing in the first version. Moreover, this comment is provocative and should probably be avoided.
Q:
Therefore, I believe the Python code, or at least the pseudo-code, is required in order to demonstrate the accuracy of their model.
A: Providing the code is sometimes not easy if there are industrial collaborations behind, like in our case. The Journal itself does not require computer codes. Neither the Special Issue does, since none of the 6 papers published in the special issue provide them.
What the code does is - once defined the variables - to pass the constraints to the gurobi solver one by one, which is just a straightforward list of commands. Loops are defined by the multiplicity of the constraints on the right, while the constraints are written according to what is written on the left.
An example for constraint (6) follows:
for b in range(len(B)):
for i in range(len(V)):
if i in C[b]:
model.addConstr(st[b,i]+ p[b][i]<=ht[b,i])
Round 3
Reviewer 1 Report
It is my responsibility as a reviewer to ensure that the methodology proposed is accurate. Using a MIP model (line 325-346) to prove the other MIP model (line 245-258) is grotesque. The decision variable, qbi (as listed in Table 1), is even not used in the proposed model (line 245-258).